# DSM Extraction Based on Gaofen-6 Satellite High-Resolution Cross-Track Images with Wide Field of View

**DOI:** 10.3390/s23073497

**Published:** 2023-03-27

**Authors:** Suqin Yin, Ying Zhu, Hanyu Hong, Tingting Yang, Yi Chen, Yi Tian

**Affiliations:** Hubei Key Laboratory of Optical Information and Pattern Recognition, School of Electrical and Information Engineering, Wuhan Institute of Technology, Wuhan 430205, China; yinsuqin@stu.wit.edu.cn (S.Y.); hhyhong@wit.edu.cn (H.H.); yangtingting@stu.wit.cn (T.Y.); chenyi@stu.wit.edu.cn (Y.C.); 04004069@wit.edu.cn (Y.T.)

**Keywords:** GF-6 satellite, DSM, wide-field high-resolution camera, rational function model, projection trajectory method, semi-global block matching

## Abstract

Digital Surface Model (DSM) is a three-dimensional model presenting the elevation of the Earth’s surface, which can be obtained by the along-track or cross-track stereo images of optical satellites. This paper investigates the DSM extraction method using Gaofen-6 (GF-6) high-resolution (HR) cross-track images with a wide field of view (WFV). To guarantee the elevation accuracy, the relationship between the intersection angle and the overlap of the cross-track images was analyzed. Cross-track images with 20–40% overlaps could be selected to conduct DSM extraction. First, the rational function model (RFM) based on error compensation was used to realize the accurate orientation of the image. Then, the disparity map was generated based on the semi-global block matching (SGBM) algorithm with epipolar constraint. Finally, the DSM was generated by forward intersection. The GF-6 HR cross-track images with about 30% overlap located in Taian, Shandong Province, China, were used for DSM extraction. The results show that the mountainous surface elevation features were retained completely, and the details, such as houses and roads, were presented in valleys and urban areas. The root mean square error (RMSE) of the extracted DSM could reach 6.303 m, 12.879 m, 14.929 m, and 19.043 m in valley, ridge, urban, and peak areas, respectively. The results indicate that the GF-6 HR cross-track images with a certain overlap can be used to extract a DSM to enhance its application in land cover monitoring.

## 1. Introduction

The digital surface model (DSM) represents a three-dimensional model of the undulating state of the Earth’s surface and all objects (buildings, forests, etc.) above the Earth’s surface [1]. The DSM plays a crucial role in natural resources and urban management, such as the monitoring and management of urban development status, forest growth, the analysis of geological disasters, etc. [2,3,4]. DSM can be generated from stereo optical images obtained by air-based and space-based platforms for different accuracy requirements [5]. DSM based on satellite platforms has much greater coverage than air-based platforms, which has become an effective way to describe surface elevation on a large scale.

For optical remote sensing satellites, stereo images can be obtained in multiple ways, including single-satellite multi-linear array imaging mode, single-satellite single-line array agile imaging mode, and video satellite area-array starring imaging mode [6]. The advantages and disadvantages of these methods are shown in Table 1. For instance, the System Probatoire d’Observation dela Terre-5 (SPOT-5) satellite, launched by Centre National d’Etudes Spatiales (CNES) on 4 May 2002, has the ability of both in-orbit and cross-track stereo imaging [7]; it carried an optical high-resolution geometric (HRG) camera with 5 m ground sampling distance (GSD) for cross-track and a high-resolution stereoscopic (HRS) camera with 10 m GSD for in-orbit, and the general elevation errors of DSM are 6.5 m and 5.2 m for HRG and HRS [8], respectively. The CartoSAT-1 (IRS P5) satellite of India was launched in 2005, which adopted a two-line array stereo mapping model with a spatial resolution of 2.5 m [9]. The average lateral error of the extracted DSM is 6.7 m, and the average vertical accuracy is 5.1 m [10]. China’s Gaofen-7 (GF-7) satellite was launched in 2019 and had a dual-line array high-resolution camera with a spatial resolution of 0.65 m that can be used for 1∶10,000 stereo mapping [11,12]. The Advanced Land Observation Satellite (ALOS) was launched in 2006 by the Japan Aerospace Exploration Agency (JAXA); it was equipped with a three-line array stereo mapping sensor to achieve a ground pixel resolution of 2.5 m [13,14], and the accuracy of DSM can be better than 5 m [15]. The ZiYuan-3 (ZY-3) satellite, the first Chinese civilian stereo mapping satellite launched in 2012, carried a three-line array camera with a spatial resolution of 2.1 m for the nadir camera and 3.5 m for the forward and backward cameras [16,17], which was mainly used for 1∶50,000 stereo mapping. Tang et al. revealed that the vertical accuracy of DSM can be better than 2 m [18].

With the development of high-resolution (HR), high-stability, and high-agility platform technology, high-resolution agile satellites can capture images with single-line array. This can be divided into two groups: one-track stereo imaging and cross-track (side swing) stereo imaging. When relying on the adjustment of the satellite imaging pitch and yaw attitude angle during the transit of one-track imaging, rendezvous imaging is carried out along the orbit direction to achieve one-track stereo observation. In different orbits, the same area is imaged in transit by adjusting the side track and yaw angle of the satellite, and rendezvous imaging is carried out in the vertical direction, so as to realize the stereoscopic observation for cross-track. The major satellites with one-track imaging are as follows: the Worldview-1 [19], Worldview-2 [20], Pleiades-1 [21], and GeoEye-1 [22] satellites were launched successively with a spatial resolution of 0.5 m or better from the years 2007 to 2009. The accuracy of DSM is at the level of 3 m for Worldview-1 [23], and based on Worldview-2 stereo images, extracted DSM can achieve a median error of less than 1.9 m in forested areas [24]; the DSM extracted from GeoEye-1 stereo pair attained better vertical accuracy (2.04 m) [25]. The RMSE of DSM using Pleiades-1 stereo pairs is 1.17 m in urban areas [26]. Similarly, the major cross-track satellites are as follows: in 1999, the world’s first high-resolution commercial satellite IKONOS was successfully launched, with a spatial resolution of 1 m. It uses the attitude maneuver capability of the satellite platform to achieve multi-angle imaging in the same and different orbits. The root mean square error (RMSE) of DSM in the area, excluding trees, can be 2–3 m, and in the bare ground, the accuracy can be better than 1 m [27]. Subsequently, the Quickbird satellite, successfully launched in 2001, provided a ground resolution of 0.61 m in panchromatic and 2.4 m multiband, and for certain lakes and bare soil, the elevation error of 0.5 and 1.3 m can be obtained, respectively [28].

Video satellite area-array starring imaging mode can also obtain stereo images; DSM and DEM products of the Jilin-1 video satellite have been tested by several domestic authorities who found that the relative elevation accuracy is better than 3 m when the resolution is 2 m [29]. The elevation accuracy of DEM data extracted using SkySat 1 and 2 is 1–2 m relative and 2–3 m height accuracy [30].

Increasing the imaging swath is one of the ways to improve the efficiency of image acquisition for high-resolution satellites. The Wide Field of View (WFV) imager has very large field angles. The Gaofen-1 (GF-1) WFV images, with a total swath of 800 km in a spatial resolution of 16 m, were used for stereo mapping. The elevation accuracy of DSM extracted from the GF-1 WFV images is 30 m [31].

Gaofen-6 (GF-6) was successfully launched in 2018 and is another wide-field optical satellite of a high-resolution Earth observation system in China after GF-1 [32]. The satellite’s orbit height is about 645 km. The GF-6 satellite is primarily used for forest and agricultural resource monitoring. There are two primary imaging payloads, a high-resolution (HR) camera, and a WFV camera. The basic parameters of the GF-6 satellite are illustrated in Table 2. The HR camera can obtain panchromatic (PAN) images with a spatial resolution of 2 m and multispectral (MS) images of 8 m [33]. GF-6 satellite has cross-track (no side swing) imaging, and the swath of the HR camera is about 90 km, which is wider than 30–40 km than that of the other satellites with the same GSD. The field of view of the ordinary HR satellite is commonly 3–4 degrees. Nevertheless, the field of view of the HR camera is 8.6 degrees. As shown in Figure 1, the intersection angle of the overlapping area in the cross-orbits can reach about 10 degrees. Therefore, the GF-6 satellite HR camera has the potential for 3D information extraction.

The challenge of extracting DSM from the GF-6 WFV satellite cross-track images is that the intersection is not as large as the common stereo images by the multi-linear arrays or agile imaging. However, the intersection accuracy of the stereo image is directly related to the intersection angle when the matching error of the corresponding image points is fixed. Generally speaking, the larger the intersection angle, the higher the intersection accuracy. When the intersection angle is fixed, the matching method with high stability and accuracy should be selected insofar as possible. As the cross-track images of GF-6 are usually captured at different times, the differences between cross-track images are affected by the imaging condition and land cover changes caused by the season, weather, etc., which will affect the matching accuracy and leads to the intersection error. To guarantee the elevation accuracy, it is necessary to select image pairs with an appropriate overlap to ensure enough intersection angle and short imaging time intervals and avoid the big differences between two images.

Besides the data selection, image matching is a crucial step of the DSM extraction for GF-6 cross-track images. In the field of computer vision, the semi-global matching (SGM) algorithm proposed by Hirschmuller and Scharstein in 1999 is the mainstream matching algorithm for three-dimensional information extraction, based on an energy-minimizing stereo matching algorithm to obtain the disparity map [34]. SGM has been used to extract the DSM for remote sensing stereo images and also integrated into famous commercial software ENVI and the ERDAS space image DSM production module [35]. However, SGM does not perform well in shadow, low texture, repeated texture, and disparity jump regions. Thus, the semi-global block matching (SGBM) algorithm was proposed in 2008, based on the SGM algorithm, which uses block matching and pixel-cost aggregation techniques to improve the efficiency and accuracy of the algorithm when calculating the disparity map [36]. In recent years, deep-learning-based matching methods have been developed, including distributed methods and end-to-end methods [37]. At present, most of the deep learning intensive matching network models rely on training tags, which limits the universality.

In this paper, the relationship between the intersection angle and the overlap of the cross-track images is analyzed. Based on data selection for DSM extraction, the error-compensated rational function model (RFM) is used to eliminate the systematic error of the cross-track stereo images. The SGBM algorithm is used to obtain the disparity map with the epipolar constraint. DSM is obtained by forward intersection. A couple of the GF-6 panchromatic images with about 30% overlap were selected to conduct the experiments. The results of different areas, including urban, valleys, ridges, and mountain peaks, were analyzed by both qualitative and quantitative evaluation.

## 2. Materials and Methods

In this paper, the DSM extraction method for WFV HR camera cross-track images mainly consists of four steps: (1) stereo image orientation based on a rational function model; (2) epipolar image generation using the projection trajectory method; (3) dense matching based on SGBM; and (4) DSM generation. The flowchart of DSM extraction using GF-6 satellite images is illustrated in Figure 2.

### 2.1. Accurate Orientation of Stereo Images Based on an Error-Compensated RPCs Model

The rational polynomial coefficients (RPCs) model is currently the most effective and simplest form of geolocation alternative model for high-resolution satellite image processing, which is commonly used in HR satellite image processing and applications [38,39]. The RPCs model depicts the relationship between geodetic coordinates (longitude, latitude, and elevation) and their corresponding image point coordinates (row numbers and column numbers) with the following equation:(1)$$\left\{\begin{array}{l} r_{n}=\frac{N_{1}(X_{n},Y_{n},Z_{n})}{N_{2}(X_{n},Y_{n},Z_{n})}\\ c_{n}=\frac{N_{3}(X_{n},Y_{n},Z_{n})}{N_{4}(X_{n},Y_{n},Z_{n})} \end{array}\right.$$
where $\left(X_{n},Y_{n},Z_{n}\right)$ and $\left(r_{n},c_{n}\right)$ represent the regularized coordinates of ground coordinates (X,Y,Z) and image point coordinates (r,c) after translation and scaling, respectively, and the values are [−1, 1] to ensure the stability of the calculation; $N_{i}(i=1,2,3,4)$ is a three-order polynomial.

The error effect of the satellite position and the attitude angle of the sensor will lead to a certain systematic error in the rational function model established according to these parameters; these errors will affect the RPC parameters, hence affecting the geometric positioning accuracy of the image. Fraser et al. [40] demonstrate the utility of utilizing bias-compensated RPCs for high-accuracy localization in high-resolution satellite images. The transformation parameters are computed by a certain number of GCPs, and a strictly rational function model is established [41]. This method needs to solve the transformation relationship between the measured coordinates of image points and the image coordinates calculated by the RPCs model. Generally, the affine transformation is used to correct the existing system error to correct the error in the row direction and the error in the column direction. The affine transformation model [42] is as follows:(2)$$\left\{\begin{array}{l} \Delta E_{x}=e_{0}+e_{1}\cdot r+e_{2}\cdot c-x\\ \Delta E_{y}=f_{0}+f_{1}\cdot r+f_{2}\cdot c-y \end{array}\right.$$
where $\Delta E_{x}$ and $\Delta E_{y}$ denote the error correction in the direction of rows and columns, $\left(e_{0},e_{1},e_{2},f_{0},f_{1},f_{2}\right)$ denotes the error compensation parameter of the affine transformation system, x and y denote the measured coordinates of GCPs on the image, and r and c denote the projection value of the horizontal and vertical coordinates of the GCPs projected to the image surface using RPC parameters.

From Equation (2), the error equation of each tie point and the GCP can be obtained [43], and then the strategy of direct column modification of the equation is used to eliminate the unknowns of the ground coordinates of a class of tie points. Eventually, the precise RPC model is gained using the traditional beam adjustment method, which improves the geometric positioning accuracy of the image.

### 2.2. Epipolar Image Generation Based on the Projection Trajectory Method

Due to the large width of the remote sensing images, the determination of the corresponding point between the left image and the right image is time-consuming when using the violent search method pixel by pixel without any constraint conditions. Furthermore, in complex situations, such as weak texture or repetition texture, it is complicated to find the exact corresponding point in the area. In order to improve the search efficiency and matching accuracy, the stereo images need to conduct epipolar rectification before image matching, so that the search space of image matching is reduced from a two-dimensional to a one-dimensional search with epipolar constraint, which means the corresponding points are located on the corresponding epipolar lines [44].

In this paper, the image after precise orientation is divided into blocks, which can lower the error of generating epipolar images. Because the epipolar line of the satellite image is approximately linear and locally conjugated [45], the epipolar image pairs are generated using the projected trajectory method. First, we select the coordinates of two image points in the left image, as point a and point b, and the inverse solution of the opposite object coordinates based on the refined RPC model of the left image, as point A and point B. Then, the inverse solution of the opposite image coordinates is based on the refined RPC model of the right image, as point c and point d. Based on the above, we can then find the approximate linear equation of the same name for the epipolar lines, which are line ab and line cd. The epipolar lines of the same name corresponding to each line on the left and right epipolar images are then calculated in turn. Finally, epipolar lines were resampled to obtain accurate epipolar images [46].

### 2.3. Dense Image Matching

The SGBM algorithm [47] is used to generate disparity maps of epipolar images, which are divided into four steps: (1) gradient information of images based on pre-processing; (2) the calculation of cost is used to evaluate the similarity of the two image blocks; (3) dynamic planning is used to find an optimal matching path on each pair epipolar, which obtain the optimal disparity when the global energy function is minimized; and (4) post-processing is used to obtain optimized disparity maps.

#### 2.3.1. Pre-Processing of Image

The Sobel operator is used to process the left and right epipolar images, the processed images are mapped into new image pairs [48]. The gradient calculation formula of the Sobel operator in the x-direction is shown in (3):(3)$$\begin{array}{l} SobelP(i,j)=2[P(i+1,j)-P(i-1,j)]+P(i+1,j-1)\\ -P(i-1,j-1)+P(i+1,j+1)-P(i-1,j+1) \end{array}$$
where the central pixel is represented by P(i,j). By multiplying the convolution kernel in the x-direction and the corresponding pixel value, the gradient value of the pixel P in the x-direction can be obtained.

The mapping function is shown in Formula (4):(4)$$P_{N}=\left\{\begin{array}{l} 0,V_{p}<-C_{P}\\ V_{p}+C_{P},-C_{P}\leq V_{p}\leq C_{P}\\ 2\cdot C_{P},V_{p}\geq C_{P} \end{array}\right.$$
where $V_{p}$ represents the pixel value, $P_{N}$ represents the pixel value of the new image after mapping, and $C_{P}$ is the cut-off value for Sobel filtering in the x-direction, which is a constant parameter.

#### 2.3.2. Calculation of Cost

After preconditioning, the gradient information of the left and right images can be obtained, which makes it easy to calculate the cost. The cost consists of two parts: one is the gradient cost obtained by sampling the gradient information of the preconditioned image, and the other is the SAD (sum of absolute difference) cost obtained by sampling the original image [49]. These two costs are calculated in the SAD window. The SAD algorithm is shown in a formula in (5):(5)$$C_{SAD}(x,y,d)=\sum_{i,j\in D}\left|I_{L}(x+i,y+j)-I_{R}(x+i+d,y+j)\right|$$

In Equation (5), assuming that the pixel of the left image to be matched is p and the disparity value of this point is d, the supporting window is D. The SAD window with the size of ${\left(2n+1\right)}^{2}$ should be constructed to cover the left image to select all pixels in the window area, then, the same-size window should be used to attempt to cover the right image area and select all the pixels in this area. $I_{L}$ is the pixel value of the covered area on the left side, $I_{R}$ is the pixel value of the covered area on the right side, and the value of the left pixel minus the value of the right pixel. Lastly, the sum of absolute values of the difference between all pixels is computed according to moving the window of the right image in turn. The algorithm is used for image block matching to evaluate the similarity of the two image blocks.

#### 2.3.3. Dynamic Planning

The local block cost is computed using the method of domain summation; matching the cost of each pixel point will contain the information of the surrounding local area, and then the cost aggregation from multiple directions is considered (default to four dynamic programming paths in SGBM) [50], as shown in Formula (6):(6)$$S(p,d)=\sum_{r}L_{r}(p,d)$$
where S(p,d) is the sum of the matching costs of the pixel p in multiple paths, and $L_{r}(p,d)$ denotes the matching cost of the pixel p along the path r.

For pixel p, the accumulation function in the direction of *r* is shown in Formula (7):(7)$$L_{r}(p,d)=C(p,d)+\left\{\begin{array}{l} L_{r}(p-r,d),\\ L_{r}(p-r,d-1)r+P_{1},\\ L_{r}(p-r,d+1)+P_{1},\\ \underset{k=d_{\min},\ldots d_{\max}}{\min}L_{r}(p-r,i)+P_{2} \end{array}\right\}-\underset{k=d_{\min},\ldots d_{\max}}{\min}L_{r}(p-r,k)$$

The first item C(p,d) of the formula represents the matching cost, when the pixel point is p and the disparity is d; the second item represents the minimum value of the previous pixel cost accumulation of pixel point p on the path r; and the last item is the minimum cost value to subtract the cost in the accumulation direction. $P_{1}$, $P_{2}$ is the constant term, and the value of $P_{2}$ must be larger than $P_{1}$. In SGBM, it should be noted that if the value of $P_{2}$ is too small, there will be mismatches, then $P_{1}$ and $P_{2}$ of values are too large, resulting in an overly smooth edge. $P_{1}$ and $P_{2}$ of the mathematical models need to satisfy the equation, as shown below:(8)$$P_{i}=u\cdot cl\cdot W_{SAD}\cdot W_{SAD}$$
where cl is the number of channels of the image to be matched, $W_{SAD}$ is the size of the SAD window mentioned above, and u is a constant.

After completing the cost accumulation, the winner-take-all (WTA) algorithm is used to obtain the minimum position in the aggregation within the range of disparity search [51], which is the optimal disparity of the current pixel.

#### 2.3.4. Post-Processing of Disparity Map

After post-processing uniqueness detection, the error parallax is eliminated, and the purpose of sub-pixel interpolation is to make the disparity of object surfaces smoother, and for the error caused by occlusion, the left–right consistency check can be used for processing [52]. Lastly, optimized disparity maps can be obtained.

### 2.4. DSM Generation

Based on the optimized disparity map obtained above, the projection matrix is computed from the refined RPC model, 3D point cloud data are acquired by forwarding rendezvous, and the point cloud is filtered to eliminate outliers. The processed point cloud data are subjected to grid processing to obtain DSM.

### 2.5. Experimental Data

As GF-6 is not specially designed for surveying and mapping, the intersection angle is only relative to the overlap of the across-track images if the satellite keeps nadir imaging. The relationship between the intersection angle and the overlapping degree of the image pair and the relationship between the intersection angle and elevation error when the image matching accuracy can reach one pixel are shown in Figure 3. With the increase in stereo image overlap, the intersection angle decreases. With the increase in the intersection angle, the intersection accuracy increases. When the intersection decreases to five degrees and the overlap is about 40%, the elevation error caused by one-pixel matching deviation will be over 20 m, as shown by the black dotted line in Figure 3. It is important to select images with appropriate overlap. Considering the intersection accuracy and data utilization, we think that 20–40% overlap is the recommended range, as shown by the gray dotted line in Figure 3.

In this paper, the cross-track image pairs with 30% overlap were selected for the experiment. The area is located in Taian, Shandong Province, China, and the topography is primarily mountainous. The acquisition times of the left image and right image were 23 January 2019, and 27 January 2019, respectively. There are more clouds in the upper part of the overlapping area of the right image, which would lead to many mismatches and affect the generation of DSM. Consequently, the bottom half of the overlapping area was used as the test area to conduct the experiments of DSM generation and accuracy analysis, as it has favorable image quality and a clear texture without significant area noise. The coverage of the test images is shown in Figure 4.

The Advanced Spaceborne Thermal Emission and Reflection Radiometer Global Digital Elevation Model (ASTER GDEM) with a spatial resolution of 30 m was used as the absolute elevation reference data to evaluate the DSM elevation accuracy generated by the method in this paper. A Digital Orthophoto Map (DOM) with a spatial resolution of 2 m was also utilized for the source of adjustment control points and checkpoints.

In this paper, the commercial software ENVI 5.3 was used to extract the DSM with a resolution of 10 m, and a comparison experiment was conducted with the DSM extracted by this method to further verify the feasibility of this method.

## 3. Experiment Results and Analysis

### 3.1. Orientation Results

In this paper, 82 tie points are automatically matched, and 32 object points are evenly manually selected on different terrains, including 16 GCPs and 16 checkpoints (CKPs). Based on the error-compensated RPCs model, the affine transformation parameters of the stereo images are estimated according to the principle of least squares by using 16 GCPs. Thus, the system positioning error of the original RPC model is eliminated. The orientation accuracy of GCPs and CKPs is shown in Table 3.

According to accuracy results after image orientation in Table 3, the mean errors of the GCPs both in object and image space are virtually zero, and the mean errors of the CKPs are also minimal, which means that the systematic error of the rational function model is almost excluded after adjustment. The RMSEs of elevation for GCPs and CKPs were 6.37 m and 5.55 m, respectively, and the RMSEs for both the GCPs and CKPs in the image space reached the sub-pixel level. It should be noted that the vertical error is larger than the horizontal error. The reason for this may be that the intersection angle of the image pair of the GF-6 satellite HR camera is smaller than that of the mapping satellite.

After model orientation, the epipolar images were generated using the projection trajectory method. The vertical parallax accuracy of the generated epipolar images were able to reach the sub-pixel level, according to the corresponding points obtained by image matching using the SIFT operator.

### 3.2. DSM Results

After the epipolar image generation, the initial disparity map was generated based on SGBM, and the disparity map was optimized by a consistency check. The three-dimensional coordinates of the corresponding point pairs were generated by the forward intersection to obtain the point cloud data. After filtering and the removal of outliers, the point cloud data were rasterized to generate DSM. The DSM extracted from the experimental images retained the mountainous topography features, as shown in Figure 5.

The spatial resolution of the extracted DSM is 10 m, and the elevation ranges from 37.3 m to 1582.8 m. From an overall optical view, the DSM extracted from GF-6 panchromatic images can recover the surface morphology of the proposed method.

### 3.3. DSM Accuracy Analysis

The test area included four different geographical types, namely valleys, ridges, peaks, and flat urban areas. In order to verify the elevation accuracy of the DSM extracted from GF-6 satellite images in different terrains, we conducted a local elevation profile analysis and pixel-by-pixel accuracy analysis on different regions of the various terrains. For peak topography, the three main peaks of Mount Tai with the largest topographic reliefs were selected. For ridge topography, three areas with the most uniform distribution in the whole experiment area were selected. For valley topography, three areas were also selected in the concentrated valley area. Due to the small cover of urban areas, only one area was selected. The selected valleys, ridges, peaks, and flat urban areas are marked by purple, green, yellow, and black boxes, respectively, as shown in Figure 5. The corresponding original images are shown in Figure 6.

Figure 7, Figure 8, Figure 9 and Figure 10 show the reference DEM of the valley areas, ridge areas, peak areas, and urban areas, with the DSM extracted from the 2 m panchromatic image of the GF-6 satellite by ENVI and through the method outlined this paper. Visually, the DSM generated by this method and ENVI both retain good ground feature characteristics. In addition, more details are displayed in the urban area, such as houses, roads, and other buildings, as shown in Figure 10b,c. The DSM extracted using the method of this paper is closer to the reference DEM and smoother. The DSM generated by ENVI is very rough and has a lot of noise.

In order to further analyze the elevation accuracy of the DSM extracted by GF-6 satellite, the elevation profiles of valleys, ridges, peaks, and urban areas are shown, respectively, in Figure 11, Figure 12, Figure 13 and Figure 14. The purple dotted line indicates the same position of the reference DEM and the DSM generated by ENVI and DSM extracted in this paper. It can be seen from the elevation profile that the overall elevation trend of the DSM extracted in this paper is closer to the reference DEM. However, the DSM elevation values generated by ENVI are relatively low, and this phenomenon is more significant in mountainous areas. The average height difference in mountainous areas is about 200 m, and the minimum height difference in valley areas is about 30 m. In the method outlined in this paper, the maximum error value is also present in the peak area because the peak area is affected by light and shadow, resulting in mismatches and large intersection errors.

The elevation accuracy of DSM extracted from the GF-6 satellite WFV HR cross-track images was further quantitatively evaluated. The ten different regions of four geographic types, including valleys, ridges, peaks, and urban areas, were selected to calculate the mean error, absolute mean error, and RMSE. The results are shown in Table 4. From the results, the RMSE of the DSM of this paper is obviously better than that of ENVI. Specifically, the RMSE of the valley area is the smallest and the accuracy is the highest, which is able to reach 6.303 m. The RMSEs of the ridges and urban areas are 12.879 m and 14.929 m, respectively. The RMSE of the mountain area is able to achieve 19.043 m, which is mainly due to the large matching error in the mountain shadow area.

## 4. Conclusions

In this paper, the DSM extraction method was presented using cross-track images of the GF-6 satellite HR camera with WFV. To guarantee the elevation accuracy, the relationship between the intersection angle and the overlap of the cross-track images was analyzed. Considering the intersection accuracy and data utilization, cross-track images with 20–40% overlap are recommended. Based on data selection, the adjustment based on the error-compensated RFM was carried out to eliminate the relative and absolute positioning errors of stereo images. The RMSEs of elevation accuracy of both GCPs and CKPs are about 6 m in object space and reach the sub-pixel level in image space. Then, the epipolar image with sub-pixel vertical parallax accuracy was generated based on the projection trajectory method. The disparity map was generated based on the SGBM algorithm. Finally, the three-dimensional coordinates of densely matched homophonous points were calculated through the forward intersection to extract DSM. The experiments revealed that the DSM generated by this paper retains complete geographical features, and the RMSE of the extracted DSM is able to reach 6.303 m, 12.879 m, 14.929 m, and 19.043 m in the valley, ridge, urban and peak areas, respectively, which means that the GF-6 satellite can provide 3D geolocation information for resource survey applications.

Considering that the intersection angle of the cross-track images of the GF6 HR camera limits elevation accuracy, some external data can be used to increase the elevation constraint, such as the laser points of the ICESat-2 satellite. The ICESat-2 is equipped with an advanced topographic laser altimeter system (ATLAS). The accuracy of the laser altimeter was 0.2 m in the plain area and 2.0 m in the mountainous area. Its observation range covers the global land, which can be used as basic data for the high-precision global ground elevation reference [53,54]. In future work, the globe laser points can be used to further improve the DSM accuracy of GF-6 cross-track images.

## Figures and Tables

**Figure 1 sensors-23-03497-f001:**
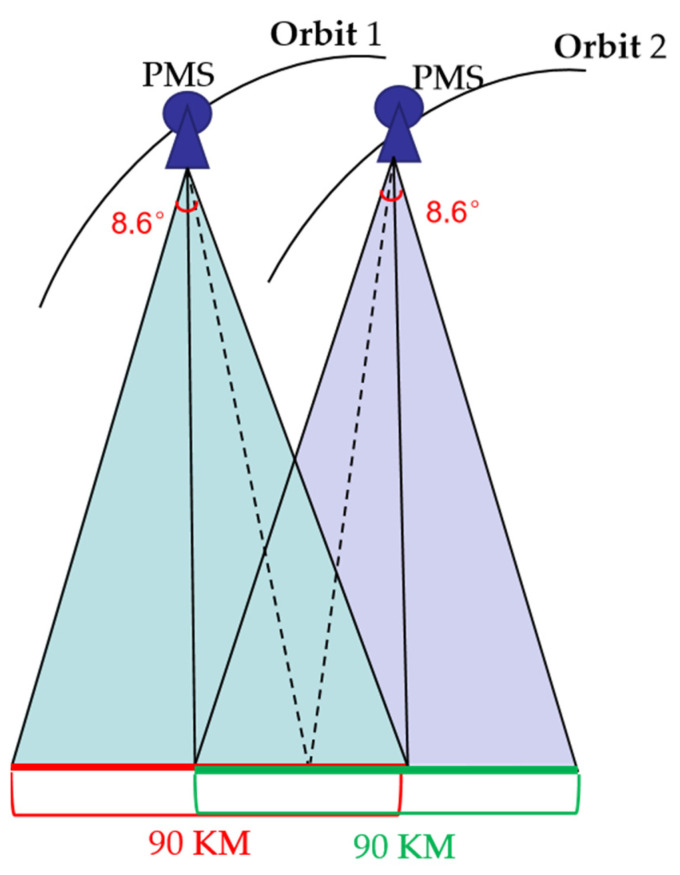
Schematic of obtaining stereo images using GF-6 high-resolution camera.

**Figure 2 sensors-23-03497-f002:**
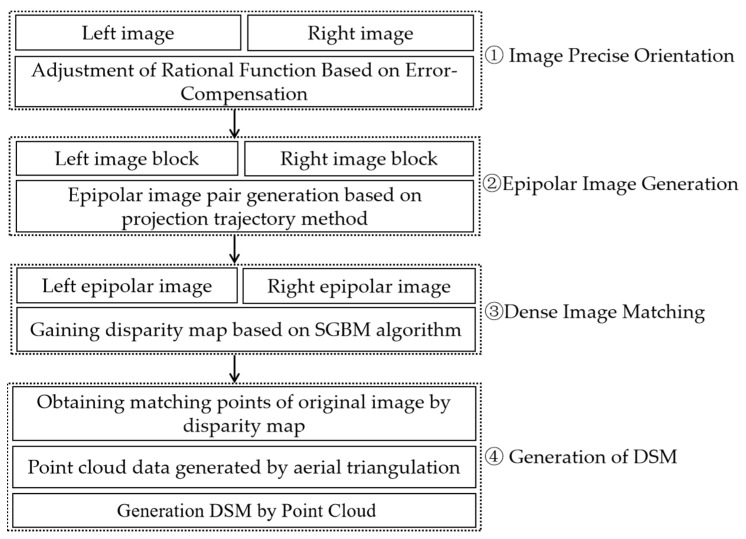
The flowchart of DSM extraction using GF-6 satellite images.

**Figure 3 sensors-23-03497-f003:**
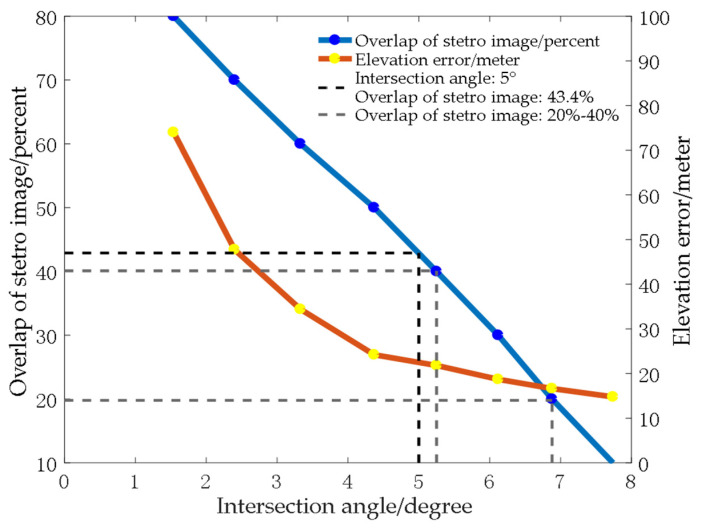
The relationship curve between the image overlaps, elevation error and intersection angle of the GF-6 satellite.

**Figure 4 sensors-23-03497-f004:**
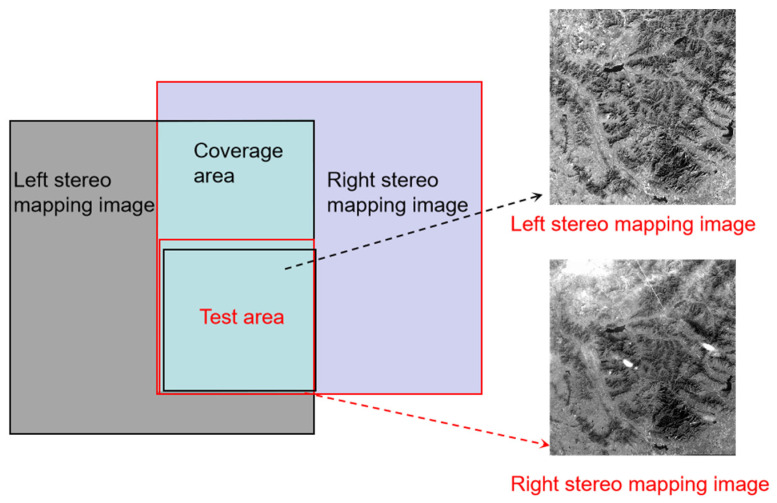
The coverage of the experimental image.

**Figure 5 sensors-23-03497-f005:**
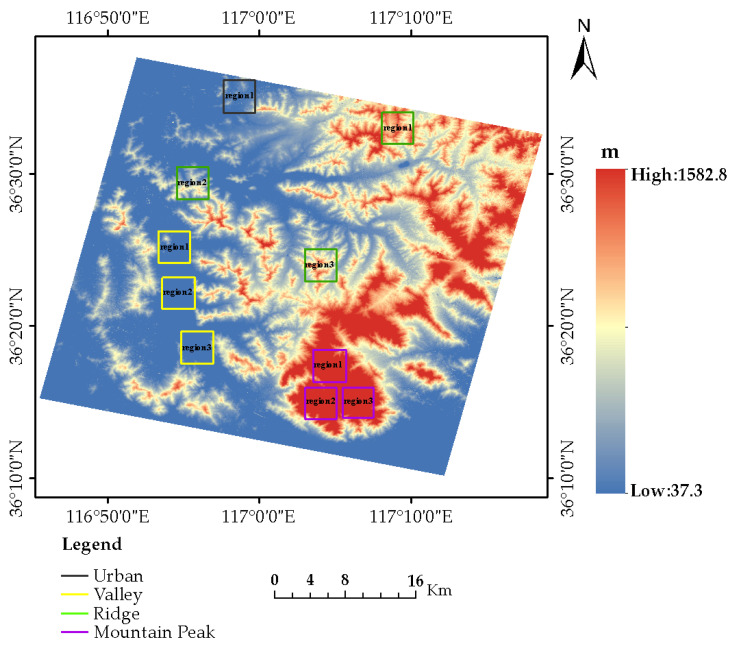
The extracted DSM from GF-6 WFV HR images.

**Figure 6 sensors-23-03497-f006:**
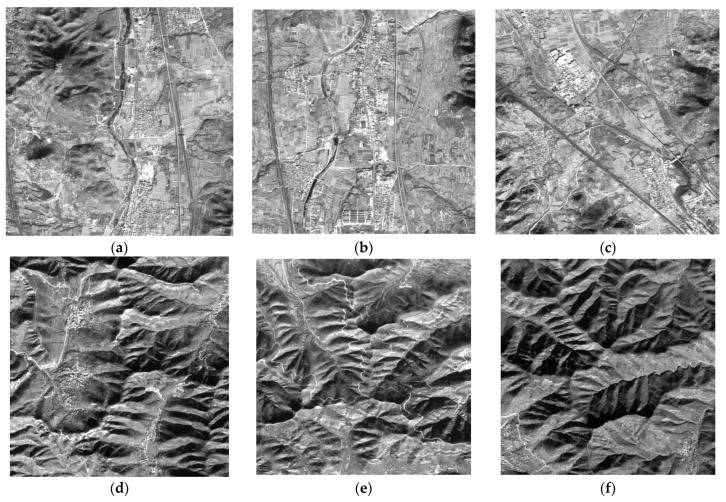
The original image of ten regions: (**a**) valley region 1; (**b**) valley region 2; (**c**) valley region 3; (**d**) ridge region 1; (**e**) ridge region 2; (**f**) region 3; (**g**) peak region 1; (**h**) peak region 2; (**i**) peak region 3; and (**j**) urban region 1.

**Figure 7 sensors-23-03497-f007:**
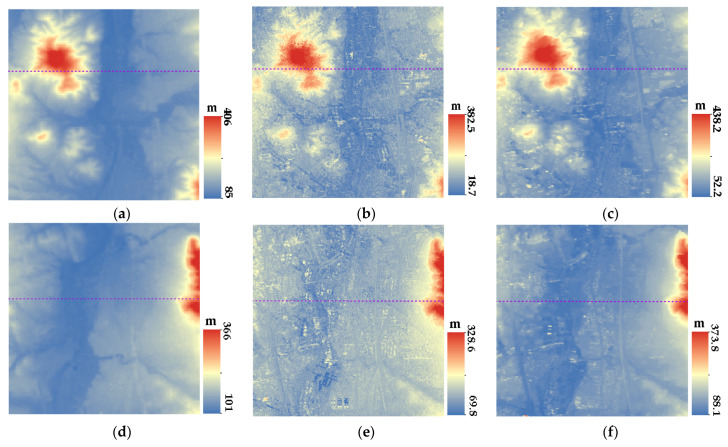
DSM of valley regions: (**a**) Reference DEM of valley region 1; (**b**) DSM extracted by ENVI of valley region 1; (**c**) DSM of this paper of valley region 1; (**d**) Reference DEM of valley region 2; (**e**) DSM extracted by ENVI of valley region 2; (**f**) DSM of this paper of valley region 2; (**g**) Reference DEM of valley region 3; (**h**) DSM extracted by ENVI of valley region 3; (**i**) and DSM of this paper of valley region 3.

**Figure 8 sensors-23-03497-f008:**
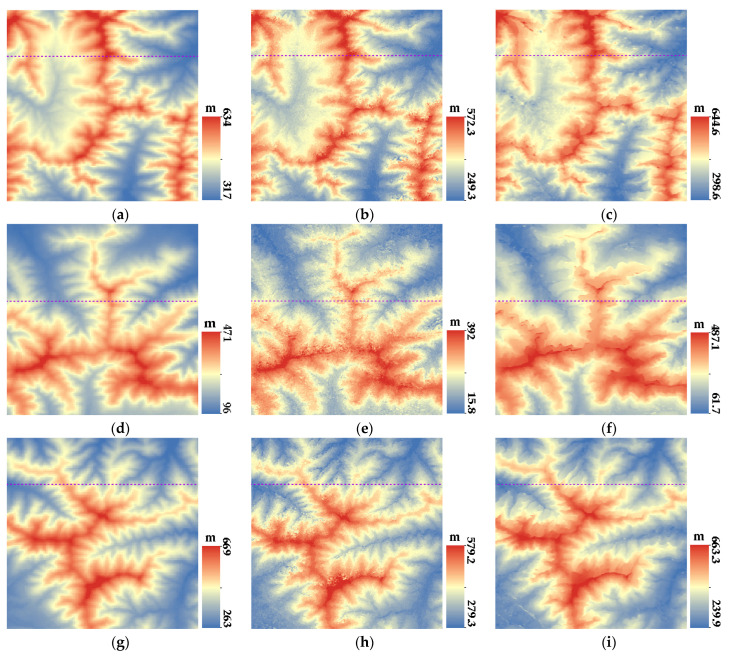
DSM of ridge regions: (**a**) Reference DEM of ridge region 1; (**b**) DSM extracted by ENVI of ridge region 1; (**c**) DSM of this paper of ridge region 1; (**d**) Reference DEM of ridge region 2; (**e**) DSM extracted by ENVI of ridge region 2; (**f**) DSM of this paper of ridge region 2; (**g**) Reference DEM of ridge region 3; (**h**) DSM extracted by ENVI of ridge region 3; and (**i**) DSM of this paper of ridge region 3.

**Figure 9 sensors-23-03497-f009:**
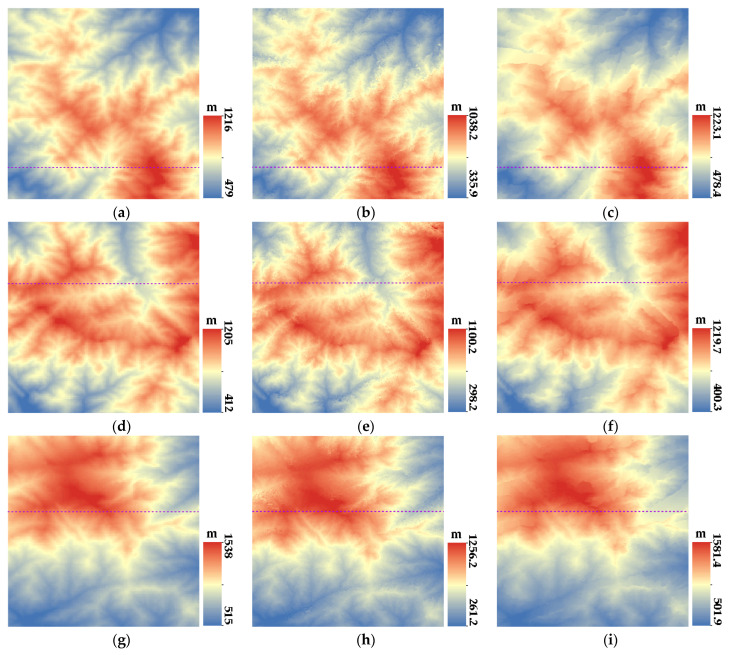
DSM of mountain peak regions: (**a**) Reference DEM of peak region 1; (**b**) DSM extracted by ENVI of peak region 1; (**c**) DSM of this paper of peak region 1; (**d**) Reference DEM of peak region 2; (**e**) DSM extracted by ENVI of peak region 2; (**f**) DSM of this paper of peak region 2; (**g**) Reference DEM of peak region 3; (**h**) DSM extracted by ENVI of peak region 3; (**i**) and DSM of this paper of peak region 3.

**Figure 10 sensors-23-03497-f010:**
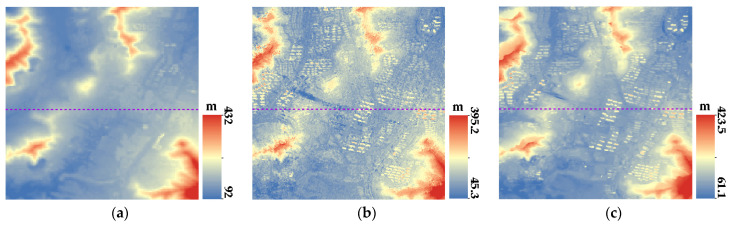
DSM of the urban region: (**a**) Reference DEM of urban region 1; (**b**) DSM extracted by ENVI of urban region 1; and (**c**) DSM of this paper of urban region 1.

**Figure 11 sensors-23-03497-f011:**
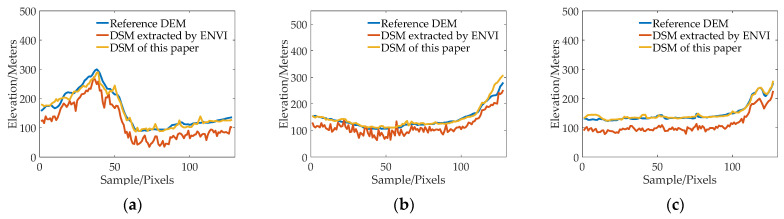
The elevation profiles comparison of three valley regions: (**a**) Region 1; (**b**) Region 2; (**c**) and Region 3.

**Figure 12 sensors-23-03497-f012:**
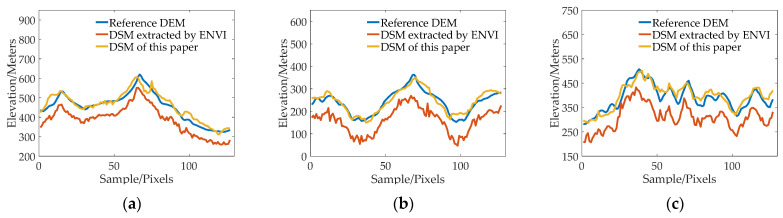
The elevation profiles comparison of three ridge regions: (**a**) Region 1; (**b**) Region 2; (**c**) and Region 3.

**Figure 13 sensors-23-03497-f013:**
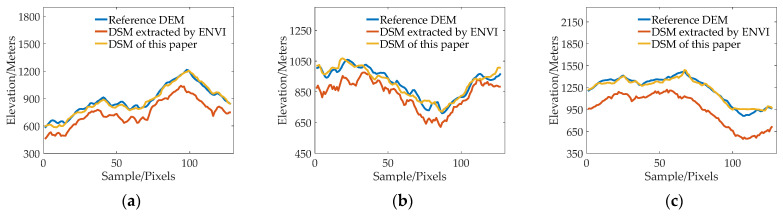
The elevation profiles comparison of three mountain peak regions: (**a**) Region 1; (**b**) Region 2; (**c**) and Region 3.

**Figure 14 sensors-23-03497-f014:**
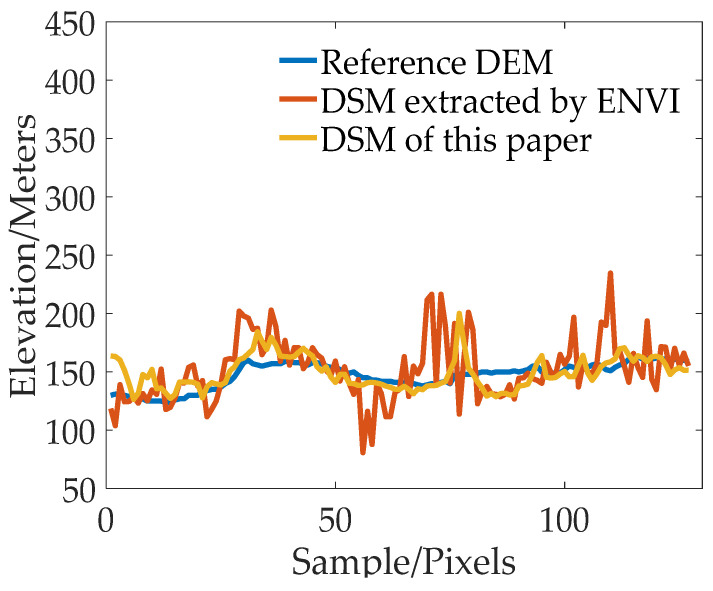
The elevation profile plot comparison of urban region.

**Table 1 sensors-23-03497-t001:** Stereo image acquisition method.

Stereo Image Acquisition Method	Advantages	Disadvantages	Applied Satellites
Single-satellite multi-linear array imaging mode	Large-scale stereo image pairs can be obtained; The attitude of the satellite is stable; It is helpful to improve the accuracy of surveying and mapping.	High cost; High requirements for satellite payloads, platforms, and launch vehicles.	Two-Line Array: SPOT-5; CartoSAT-1; GF-7. Three-Line Array: ALOS; ZiYuan-3; Tianhui-1.
Single-satellite single-line array agile imaging mode	High timeliness and flexibility of mapping; Reduced the size of the payload.	The single stereo imaging range is small; High stability of satellite attitude maneuvering is required.	One-Orbit: Worldview-1/2/3/4; GeoEye-1; Pleiades; Gaojing-1. Cross-Track: SPOT-5; IKONOS; Quickbird.
Video satellite area-array starring imaging mode	One exposure to complete the collection of all pixels, high index.	The coverage area of imaging is small.	Skysat; Jilin-1.

**Table 2 sensors-23-03497-t002:** The basic parameters of the GF-6 satellite.

Item		Technology Performance
Satellite Orbit	Orbit Type	Sun Synchronous Circle Orbit
	Average Height	644.5 km
	Revisit Period	2 Days
	Recursive Period	41 Days
High-Resolution (HR) Camera	GSD	2 m (PAN)/8 m (MS)
	Swath	≈90 km
	Field of View	≈8.6°
Wide-Field-View (WFV) Camera	GSD	<16 m
	Swath	≈800 km
	Field of View	≈64°

**Table 3 sensors-23-03497-t003:** The statistical results of the orientation of the stereo images are based on the error-compensated RPCs model.

Type		Object Accuracy/m			Image Accuracy/Pixel	
		X	Y	Z	x	y
GCPs	Mean error	−0.0012	0.0002	−0.0005	0.000	0.000
	RMSE	0.4599	0.9717	6.3701	0.269	0.459
CKPs	Mean error	−0.0697	−0.0613	0.9704	0.051	−0.088
	RMSE	0.5518	0.9013	5.5514	0.301	0.442

**Table 4 sensors-23-03497-t004:** Overall accuracy evaluation results of DSM extracted from different regions.

Region Type	Experimental Region	Elevation Accuracy of DSM Extracted by ENVI/DSM by This Paper/m		
		Mean Error	Absolute Mean Error	RMSE
Valley	Region 1	0.8116/0.148	29.990/7.981	42.917/11.087
	Region 2	0.093/0.510	19.285/6.333	33.673/8.700
	Region 3	0.273/0.462	24.777/4.588	34.575/6.303
Ridge	Region 1	0.523/0.746	77.450/9.779	63.692/12.879
	Region 2	0.533/0.113	65.763/11.087	79.323/14.150
	Region 3	0.445/0.674	65.445/10.321	78.305/13.606
Mountain peak	Region 1	0.974/0.051	113.73/14.599	141.30/20.301
	Region 2	−8.611/0.389	103.27/14.101	128.16/19.043
	Region 3	0.130/1.269	212.52/16.948	244.14/23.855
Urban	Region 1	0.326/0.105	31.125/10.594	42.281/14.929

## Data Availability

No new data were created or analyzed in this study. Data sharing does not apply to this article.

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
