# Peer review of "DSM Extraction Based on Gaofen-6 Satellite High-Resolution Cross-Track Images with Wide Field of View"

_sensors, 2023, doi:10.3390/s23073497_

Round 1

Reviewer 1 Report

The authors contribute a DSM extraction method for the cross-track stereo-images of the GF-6 satellite. The organization of this manuscript is generally good. However, I believe several issues must be addressed before this manuscript can be published in the Sensors Journal. Please see my comments below.

1.      The current abstract of the manuscript is poorly written and does not effectively convey the research's objective, methods, findings, and conclusions. To improve the abstract, the authors should follow the logical flow of 'Objective-Method-Findings-Conclusions' and focus on the most critical aspects of their research. The objective of the study should be clearly stated, and the methods used to achieve the objective should be briefly described. The research findings, including any significant results or conclusions drawn from the data, should also be highlighted. Finally, the conclusions should summarize the main findings and provide insights into the implications of the study. In particular, the abstract could benefit from a more substantial background section highlighting the significance of DSM extraction from satellite imagery and the limitations of existing methods. The study's findings should be emphasized, such as any novel techniques or improvements in accuracy achieved through the proposed DSM extraction method.

2.   The current introduction section overviews various satellite platforms and their capabilities for generating Digital Surface Models (DSM) through stereo imaging. The authors also highlight the unique features of the GF-6 satellite, such as its high-resolution camera and wider swath, which enable 3D information extraction. However, the introduction lacks a clear scientific question and justification for using GF-6 to extract DSM. To strengthen the introduction, the authors could discuss the common challenges associated with using GF-6 for DSM extraction and how their proposed solution will overcome these challenges. For example, GF-6's wider swath and the high-resolution camera can lead to large amounts of data, which can be challenging to process efficiently. Additionally, GF-6's high-resolution camera may result in more occlusions and inconsistencies in the stereo imagery, making DSM extraction more difficult. To address these challenges, the authors could propose a new DSM extraction methodology that takes advantage of GF-6's unique capabilities, such as its large field of view and high-resolution camera, while also addressing the limitations. They could also discuss how their proposed solution would perform better than existing DSM extraction methods.

3.   In Section 3.2.3, the authors do not provide a clear explanation for the selection of the test regions and why they are suitable for validating the robustness of their method in different terrain conditions. To improve this section's clarity and scientific validity, the authors should explain the criteria used to select the test regions in more detailed. They should also describe how these regions represent different terrain and environmental conditions relevant to their method's intended application. It is important to establish a scientific basis for choosing these test regions to ensure the validity of the results.

4.   The main concern is the comparison of your method with ENVI. While I appreciate the efforts made by the authors to evaluate their method using ENVI, I have some reservations about its suitability as a comparison tool for DSM extraction. ENVI was not explicitly designed for this purpose and may not be the most appropriate choice for comparison. To strengthen the validity of the evaluation, I suggest that the authors consider comparing their work with more state-of-the-art methods specifically designed for processing DSM. This would provide a more robust and meaningful comparison and enhance the impact of their research. Overall, I appreciate the authors' work and look forward to seeing how they address this concern in their revision.

Reviewer 2 Report

The study "DSM Extraction Based on GaoFen-6 Satellite High-resolution Multi-temporal Cross-track Images with Wide Field of View" is about extracting Digital Surface Model (DSM) from a Gaofen-6 satellite data. The study is well-written and focused on one objective. Also, the results show acceptable accuracy of the used data/methods. I recommend the acceptance of this manuscript after the following minor corrections.   

1.      The study title includes "High-resolution Multi-temporal Cross-track Images with Wide Field of View," which are lengthy and over-explanatory about the data used. It is suggested to include only the distinct features of the used data. Also, in the title authors used GaoFen-6; I think it's Gaofen; please double-check.

2.      Full forms of satellites shall be mentioned at their first use, e.g., SPOT, IKONOS, and other mentioned satellites.

3.      P3 L101, 102: Ordinary satellite is not an appropriate term. Please replace "ordinary" with "other".

4.      The introduction covers specific information about other satellites briefly. However, it lacks methodological discussions, e.g., what techniques are used in other DSM products or, generally, how many widely used methods are available.

5.      For better structuring, I suggest that sub-section 3.1 Experimental data shall be restructured as 2.5 Experimental data as it belongs to the methodology. Section 3 Experiment shall be restructured as 3 Results whose sub-sections are 3.1 Orientation results, 3.2 DSM results, and 3.3. DSM accuracy analysis.

6.      It is recommended to combine similar figures such as Fig 5 to 8, Fig 9-11, Fig 13-15 and 17-19 in corresponding single figures while the continuing panels (a,b,…h,i) can be used to distinguish between various types.

7.      The last paragraph of the conclusion, P16, L360 is neither concluded nor has any connection with the current study. If the suggestion of using global laser point data is required to mention, then at least it shall have a brief background in the introduction section.  

Round 2

Reviewer 1 Report

Thank you for the author's effort in revising the manuscript! I recommend publishing it in its current form in the SENSORS journal.